# The Effect of Data Augmentation Methods on Pedestrian Object Detection

**Bokun Liu [1],\*, Shaojing Su [1] and Junyu Wei [1]**

College of Intelligent Science, National University of Defense, Changsha 410003, China
\* Correspondence: liubokun16@nudt.edu.cn

**Abstract:** Night landscapes are a key area of monitoring and security as information in pictures caught on camera is not comprehensive. Data augmentation gives these limited datasets the most value. Considering night driving and dangerous events, it is important to achieve the better detection of people at night. This paper studies the impact of different data augmentation methods on target detection. For the image data collected at night under limited conditions, three different types of enhancement methods are used to verify whether they can promote pedestrian detection. This paper mainly explores supervised and unsupervised data augmentation methods with certain improvements, including multi-sample augmentation, unsupervised Generative Adversarial Network (GAN) augmentation and single-sample augmentation. It is concluded that the dataset obtained by the heterogeneous multi-sample augmentation method can optimize the target detection model, which can allow the mean average precision (mAP) of a night image to reach 0.76, and the improved Residual Convolutional GAN network, the unsupervised training model, can generate new samples with the same style, thus greatly expanding the dataset, so that the mean average precision reaches 0.854, and the single-sample enhancement of the deillumination can greatly improve the image clarity, helping improve the precision value by 0.116.

**Keywords:** infrared and visible images; data augmentation; GAN; object detection

## 1. Introduction

Night pedestrian detection has great significance to the drivers, since, because of the lack of light at night, driving vision is limited. It is difficult to distinguish pedestrian positions. Moreover, night is the peak time for dangerous events, some intruders may hide in the dark, and in most cases, situations caught by surveillance cameras are not comprehensive, due to the limitations of visible light, camera jitter and rotation, making it difficult to identify people in the dark. Visible image samples acquired under limited conditions may have problems such as low definition and sample imbalance, and insufficient sample quality can lead to poor model robustness or insufficient generalization ability. Therefore, to alleviate the above problems, data augmentation is a method worth investigating. The essence of the data augmentation method is actually to make the existing data more valuable based on the existing limited data, on the premise of not actually collecting more data. When the sample data collected is not complete enough for objective reasons, data augmentation methods can be used to generate data for new samples that are more similar to the real data distribution, and elements such as noise or random images can be introduced, so as to improve the recognition ability of the model and enhance its generalization ability. At present, data augmentation methods are very numerous. In view of images with poor lighting conditions, this paper studies the influence of different kinds of augmentation methods on pedestrian detection and discusses the accuracy and objective effect of images.

Data augmentation can be mainly divided into supervised and unsupervised data augmentation. Supervised data augmentation, that is, enhance the data on the basis of existing

data using preset data transformation rules. Supervised data augmentation mainly includes two methods: single-sample augmentation and multi-sample augmentation. Single-sample augmentation is based on image transformation, operating around the sample itself, mainly including geometric operation, color transformation, random wipe [1] and other related methods, such as shifting the image for rotation, cutting and zoom [2] and so on. In the context of infrared cameras, lidar, depth cameras and other widely used multiple sensors, multi-augmentation methods are also being increasingly applied. Multi-sample augmentation is to combine and transform multiple samples through prior knowledge to construct neighborhood values of known samples in the featured space. Wang and Perez [3] and Chawla et al. [4] proposed the SMOTE algorithm, which uses two minority class samples to synthesize new samples and adds new artificially simulated samples to the dataset, so that the minority class in the original data is no longer seriously out of balance. Tokozume et al. [5] discussed whether mixing training samples can better achieve inter-class separation based on feature visualization distributions. In 2018, Inoue H. [6] proposed an idea to improve network generalization performance through data superposition. This idea is very simple. In the paper, an efficient data augmentation method, SamplePairing, is proposed. The core idea of the algorithm is to superimpose two images randomly selected from the training set to synthesize a new sample (average pixels), using the first image label as the correct label for the synthesized image, which can increase the size of the training set from N to N × N. The model is shown in Figure 1. Images A and B are randomly grabbed from the training set. The two are basically randomly flipped and extracted to obtain the corresponding patch, and then averaged to obtain a new sample. The label is one of the original sample labels. They are then fed into the training network.

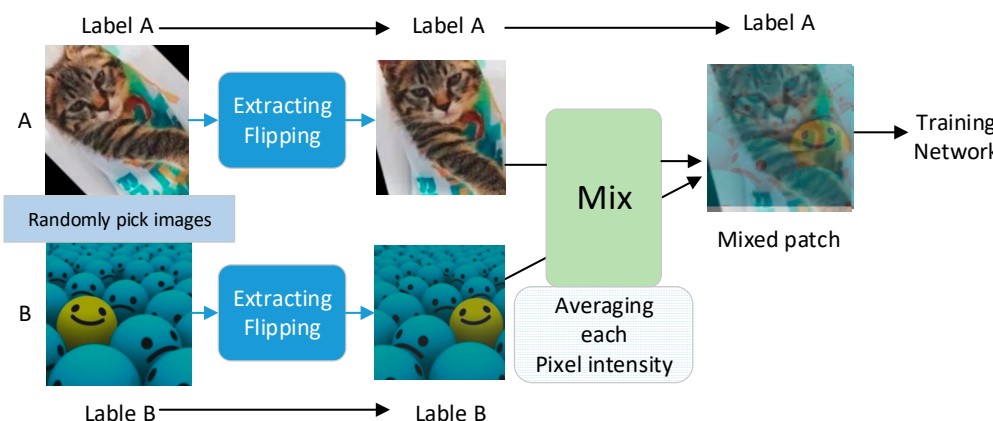

**Figure 1.** The example process of SamplePairing augmentation for cat image A and ball image B.

Unsupervised data augmentation mainly learns new images with the same distribution as the training datasets through the model and derives different augmentation methods based on the GAN network. The traditional GAN network has a generator G and a discriminator D; the generator G is used to generate samples and the discriminator D is used to judge whether the sample is a real sample. G generates fake images with random noise, and D performs binary classification training based on real and fake images. D generates a score based on the input image, which indicates whether the image generated by G is successful and further trains G to generate better images. Alec Radford [7] proposed DCGAN, which used BatchNorm and removed fully connected hidden layers for deeper architectures. After Radford et al. proposed DCGAN, many tasks in computational vision began to use generative adversarial models to solve problems faced in specific tasks. In order to enhance the robustness of the detection model, Wang et al. [8] improved the detection performance of difficult objects by automatically generating samples containing occlusion and deformation features. Subsequently, Li et al. [9] proposed a new perceptive-GAN model for perceptual generative adversarial networks, which improves small object detection by narrowing the representation difference between small objects and large

objects. The method learns high-resolution feature representations for small objects by confronting the generator and discriminator against each other.

Overall, heterologous multi-sample augmentation is bold and innovative and is simple and easy to operate. It improves the generalization ability of the network and makes the samples balanced. Unsupervised data augmentation quickly generates new samples to expand datasets without manual labeling.

However, the existing literature study has some limitations, which are summarized as follows:

a. For heterologous multi-sample augmentation, there are few studies, and the method of multi-sample stacking needs to be further enriched and improved.

b. Only a small number of researchers link the computational process of image information processing algorithms with high-level vision tasks, such as object detection, and rarely consider the feedback of high-level tasks.

c. The specific impact of different data augmentation methods on the target detection remains to be investigated.

This paper discusses the above issues and proposes some related improvements. The acronyms used in the paper are shown in the Abbreviations.

The general idea of the experiment is as the following: 1. Detect the visible light image alone and observe the model detection results without enhancement in the datasets. 2. Use infrared images to add fusion enhancement and compare the results. 3. Use the generative adversarial network to generate new images based on the existing fused images and expand the dataset and then compare the test results.

## 2. Methodology

This section introduces the main method principles of data augmentation and target detection used in the paper. We use the MSRS public dataset.

### 2.1. Supervised Image Fusion

Based on the idea of SamplePairing, the superposition of the two random pictures can effectively improve the accuracy of the model. Considering that the nighttime visible light images cannot guarantee clear imaging and it is easy to lose the effective features of the target, the infrared imaging can reflect the temperature information, supplement the visible light information, and achieve a more accurate detection of the target. Therefore, it is a good data augmentation method to superposition the images of two different light sources, which can fully realize the information complementarity and spatial correlation.

This paper introduces a task-driven fusion algorithm that combines high-level visual tasks into image spatial information processing methods to achieve true task traction. The SeAFusion algorithm [10] proposed by Tang et al. uses a semantic sensing infrared and visual image fusion framework, which can maintain the balance between low level and high level visual tasks. Under the guidance of semantic loss, the fusion images can generate more visual attraction and achieve excellent performance of advanced visual tasks. The advantage of this method is to make the fusion image evaluation criteria more comprehensive and objective; the main principle schematic is depicted in Figure 2 and it is divided into a fusion stage and segmentation stage for iterative training. First, according to the fusion network results [11], calculate spatial information loss and semantic loss, then use the combined loss to guide fusion network parameter p-step cycle update. The fusion image generated after the update obtains the new semantic loss through the segmentation network, the segmentation network parameters obtain q-step cycle update, and the new semantic loss provides feedback for the joint loss, so that the fusion network and the segmentation network iterate set times and the trained fusion network model is finally obtained [12].

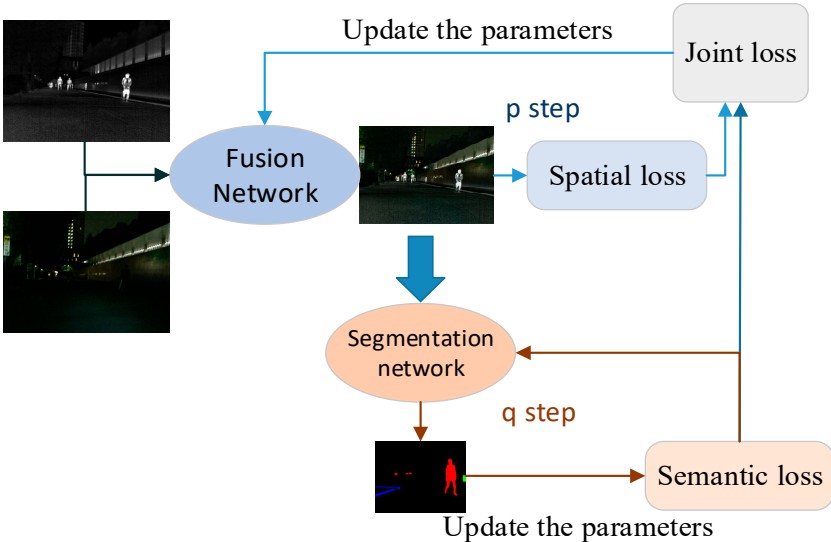

**Figure 2.** The SeAFusion algorithm training process.

The fusion network internal structure is depicted in Figure 3. The infrared and visible light images are input parallel to the feature extraction layer, and each feature extraction branch contains a common convolution layer and a dense block of gradient residuals in parallel. The common convolution layer kernel size is $3 \times 3$ and the activation function LeakyReLU enables better extraction of shallow features. Subsequent residual gradient density blocks (the RDG module) [13] extracts deep features. The fused features enter the decoder structure for feature aggregation and image reconstruction. The filling is set to be the same, the stride is set to 1, the image features are not downsampled, and the size of the fusion image is consistent with the source image, thus losing very little information.

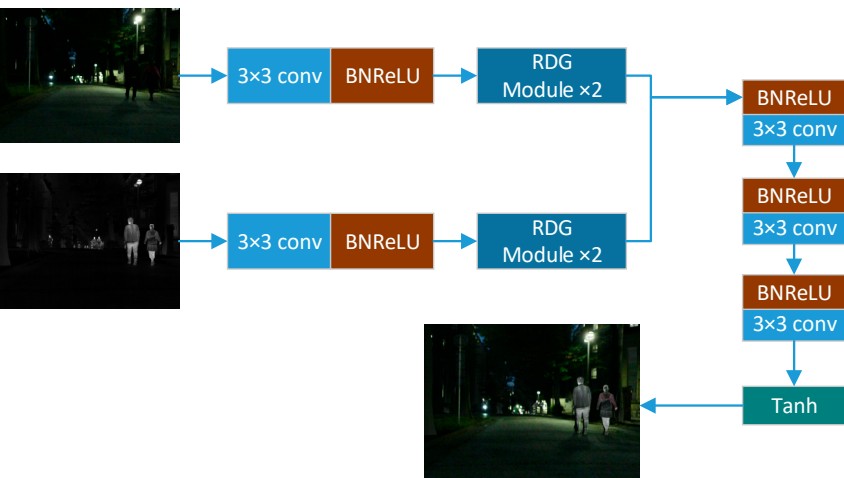

**Figure 3.** The fusion part network internal structure.

### 2.2. YOLOv5 Detection

In this paper, the YOLOv5 model was trained using our data-enhanced dataset. YOLOv5 is a single-stage target detection algorithm, mainly using CSPDarknet for feature extraction. The algorithm adds some new improvement ideas on the basis of YOLOv4, so that its speed and accuracy have been greatly improved in terms of performance. The main advantages are shown as follows:

Regarding input during the model training phase, it conducts Mosaic data enhancement, adaptive anchor calculation, and adaptive picture scaling. For benchmark networks, it incorporates some new ideas from other detection algorithms [14], which mainly include

focus structure and CSP structure. In terms Neck networks, the target detection network adds an FPN structure between the Backbone and the last Head output layer. The Head output layer has the same anchor frame mechanism as the YOLOv4 [15], the main improvement is about loss function during training, GIOU_Loss, and the DIOU_nms of the prediction box filtering.

### 2.3. Unsupervised Data Augmentation

Unsupervised representation learning is a rather deep problem in general computer vision research. A classic approach to unsupervised representation learning is to cluster the data (e. g., using K-means) and use clustering to improve the classification scores. In the context of an image, the image patches get hierarchical clustering [16] to learn a powerful image representation. Another popular method is to train autoencoders [17]; the image is encoded into a compact code and decoded to reconstruct the image as accurately as possible. These methods have also been shown to learn good feature representations from image pixels. Figure 4 shows the classic GAN network structure.

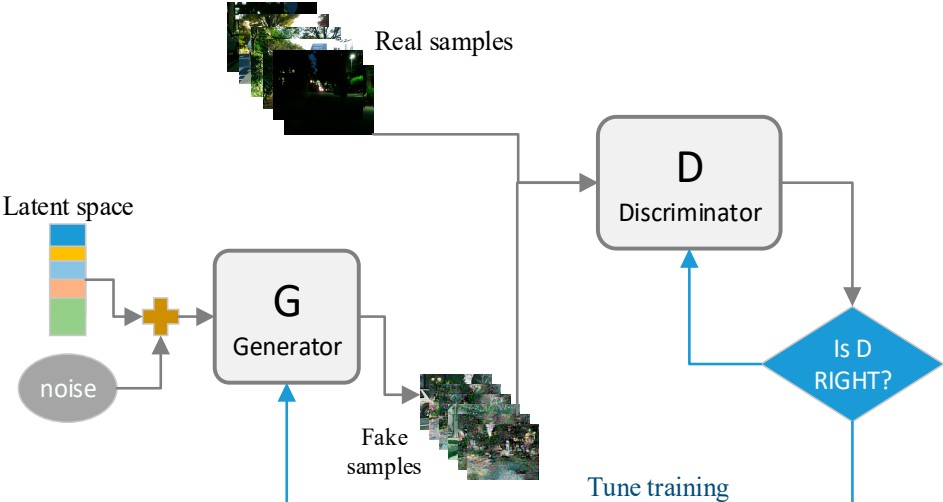

**Figure 4.** Classic GAN network structure.

GANs train the generator and the discriminator in a confrontational way. The generator is used to generate "fake" samples as realistically as possible and the discriminator is used to distinguish as accurately as possible whether the input is a real sample or a generated "fake" sample.

### 3. Improved Methods

#### 3.1. Improved Semantic Segmentation Network for Bilateral Attention

In order to link the visual task with image fusion, a semantic segmentation network is built to send feedback the fusion network when visible light and infrared images are fused. The semantic segmentation network provides the semantic loss of targets in an image, linking the high-level visual task of object detection to image fusion process [18], thus making the image data more tailored to the task requirements.

The semantic segmentation network structure is shown in Figure 5, and the feature maps of different levels are fused through two steps of feature extraction and information fusion.

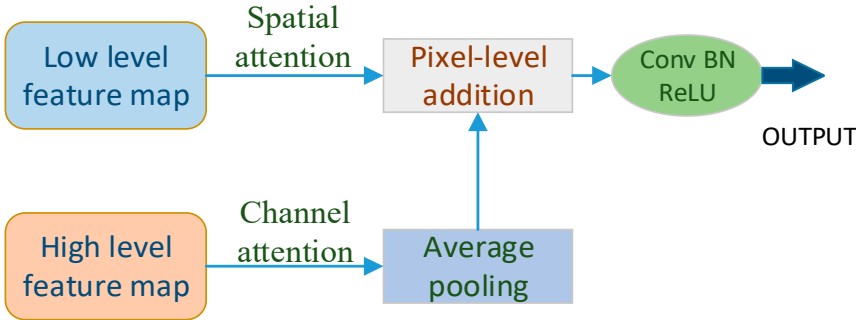

**Figure 5.** The semantic segmentation network structure.

In the first step of feature extraction, bilateral attention [19] consists of a channel attention branch and a spatial attention branch. Figure 6 shows the improved bilateral attention module. The high-level feature maps extracted by the channel attention branch can capture more precise semantic information and the low-level feature maps extracted by the spatial attention branch can capture more accurate spatial information [20]. In the second step of feature fusion, we use a novel pooling fusion block to fuse the extracted high-level and low-level feature maps. This fusion block can make full use of both high-level and low-level feature maps, and fully utilize the advantages of both so as to obtain high-quality fusion results.

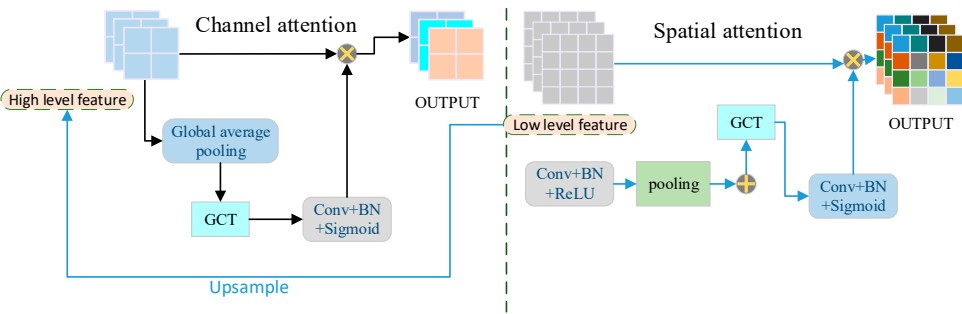

**Figure 6.** The improved bilateral attention module.

When generating the channel attention mask, the global average pooling GAP is used to reduce the dimension of the input high-level feature map and then the gated channel attention mechanism GCT module [21] is added to enhance the representation ability of the convolutional layer. After connecting to the convolutional layer, BN layer, and activation layer, the feature map is adjusted comprehensively and deeply. When generating the spatial attention mask, first we reduce the number of low-level feature channels, then combine the results of max pooling and average pooling to fully extract the spatial details of the image, then input the feature map into the GCT module, and then go through the conv + BN + Sigmoid layer so that the channels are compressed into one.

### 3.2. Residual Convolutional GAN

In the unsupervised data enhancement method, due to the use of deconvolution in the generator of DCGAN, the stride of the deconvolution collocation is more than 1, resulting in the convolution unable to isotropically cover the entire picture. Therefore, the interleaving effect appears, resulting in the "Checkerboard effect", which restricts the upper limit of the generation ability of DCGAN. Thus, this paper makes some improvements to the structure of the DCGAN generative adversarial network and makes it a Residual Convolutional GAN.

Among these changes, a resnet structure is primarily added to solve the gradient dispersion problem and the residual network enables the model network layers to be deepened. The overall framework is shown in the Figure 7.

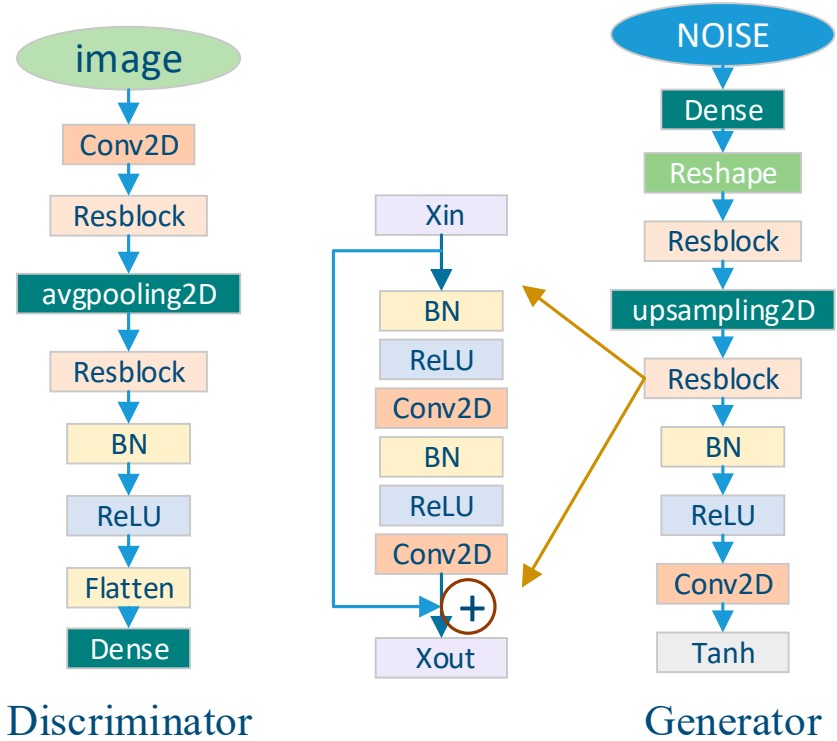

**Figure 7.** Residual Convolutional GAN structure.

The changes mainly include:

1. Removing the deconvolution of both the discriminator and the generator;
2. Building residual blocks between the convolutional layers;
3. Using AvgPooling2D and UpSampling2D to sample up or down, adjusting the length and width of the image;
4. Increasing the number of convolution layers of ResBlock to increase the nonlinear capability and depth of the network.

The inputs are the fused images, which are put into the improved GAN network to generate new samples, thereby enhancing the multi-sample enhanced datasets again. First, the generator adds random noise and enhances feature transfer through denseblock. The number of parameters are reduced. Images go through the reshape layer, ResBlock, upsampling layer, and ResBlock and batch normalization. Finally, after the activation function, false images are obtained.

### 3.3. Retinex Deillumination Algorithm

For the single-sample traditional enhancement method, it mainly adjusts the contrast and sharpness of the image. According to the characteristics of night images, this paper chooses a deillumination algorithm to remove part of the influence of the darkness and "replenish light" on the image. The Retinex deillumination algorithm select pixels layer by layer using an image pyramid. The iterative operation is performed by taking points, comparing, and averaging.

The Retinex theory [18] believes that the observed image S is obtained by reflecting the incident light L from the surface of the object and the reflectivity R is determined by the object itself and does not change with the incident light L. The relationship is seen in (1).

$$S = R \times L \tag{1}$$

where L represents the imaging part of the incident light, which directly determines the dynamic range that the pixels in the image can achieve; R represents the reflection property image of the object, that is, the intrinsic property of the image; and S represents the reflected

light image received by the human eye. Therefore, the influence of L in the original image should be eliminated, so as to preserve the reflection attribute image of the essence of the object as much as possible.

The concept of an image pyramid is simple, which is to downsample an image to represent the image in a multi-resolution form. Figure 8 shows the image pyramid. The topmost image has the lowest resolution, and the bottom layer has the highest resolution [19].

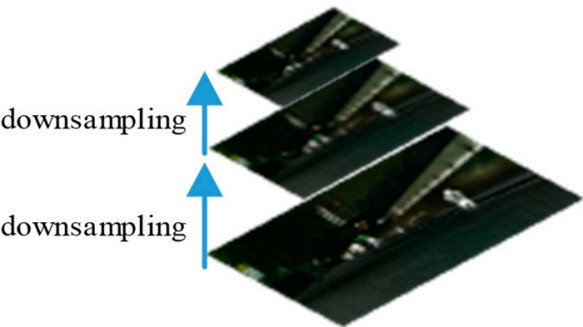

**Figure 8.** Image pyramid sketch map.

The Retinex algorithm starts from the top layer and compares each pixel with its eight adjacent pixels to estimate the reflectance component R; after the calculation of the previous layer is completed, the estimated reflectance classification is interpolated, so that the upper image of the estimated result R of one layer is the same size as the image of the next layer of the pyramid, and the same comparison operation is performed again; finally, the final result can be obtained after the eight-neighbor comparison of the original image is completed, that is, the enhanced image. Here let $S_1, S_2, \ldots, S_m$ be the points on the path, arranged from far to near, where Rc represents the final reflectivity estimate of the center position. Meanwhile, Sm is decomposed according to the Retinex theory.

The main steps are as follows:

1. Transform the original image to the logarithmic domain S (x,y);
2. Calculate the number of image pyramid layers; initialize the constant image matrix $R_0$ (x,y), which is used as the initial value for the iterative operation;
3. Eight-neighborhood comparison operations are performed from the top layer to the last layer, and the pixels on the path are calculated according to the following Formula (2):

$$R_c = \frac{S_c - S_m}{2} + \frac{R_c + R_m}{2} \tag{2}$$

$$S_m = r_m + l_m \tag{3}$$

$$R = R_0 + \frac{r_c - r_m}{2} + \frac{r_c - r_{m-1}}{4} + \cdots + \frac{r_c - r_1}{2^m} \tag{4}$$

4. Afterthe operation of the nth layer is completed, interpolate the operation result R of the nth layer to become twice the original, which is the same as the size of the n + 1 layer. When the bottom layer is calculated, the R obtained is the final enhanced image. Sc is the center point. (3) is the Logarithmic form of (1). After transformed, the intrinsic property R is calculated by (4).

## 4. Results and Discussions

### 4.1. Experimental Data

To examine the impact of the three data augmentation methods studied in this paper on object detection, we conducted extensive experiments on the MSRS dataset. The MSRS dataset is a road scene dataset, in which the training set contains more than a thousand pairs of infrared and visible light images, including day and night conditions, and we calculated the proportion of dark pixels and selected images with 75% dark pixels as poor lighting

images for testing and verification [22]. This dataset provides semantic labels for objects such as cars, people, bicycles, curves, car stops, guardrails, color tints, and backgrounds. In this paper, the common active object pedestrian is used as the detection target, and only the pedestrians are marked with the target detection frame. We choose 800 pairs of the nighttime images to train, 200 images to test, and 100 other images to validate.

*4.2. Training Configuration*

In this paper, we separately used supervised single-sample and multi-sample data augmentation, unsupervised GAN networks to generate new samples, and then compare the detection results of the new dataset in the YOLOv5 architecture. The overall experimental step is to first perform multi-sample fusion augmentation, compare the target detection accuracy. Then, the dataset of mixed fusion images is amplified by GAN to compare the accuracy of target detection [23]. Finally, the new dataset is deilluminated and enhanced to observe the detection effect and image effect.

4.2.1. Heterogeneous Multi-Sample Augmentation Training

The fusion network and segmentation network are iteratively trained, and the iteration is set to M. First, the Adam algorithm is used to optimize all parameters in the fusion network using the optimizer under the feedback of the joint loss and dynamically adjust the hyperparameter $\beta$ of the joint loss by iteration.

$$\beta = \gamma \times (m - 1) \tag{5}$$

In Equation (5), m represents the number of iterations, $\gamma$ is a constant that balances semantic loss and content loss, and the parameters of the segmentation model are updated by optimizing the semantic loss. In each iteration, the training steps for fusion model and segmentation model are p and q, respectively. The training process is shown in Table 1. The fusion network and segmentation network are iteratively trained according to a joint low-level and high-level adaptive training strategy. The flow chart is shown in Table 2. All parameters in our joint adaptive training strategy are set as follows: M = 4, p = 1500, q = 10,000, and $\gamma$ = 1. Meanwhile, our fusion model is optimized under the guidance of joint loss. Furthermore, we optimize the segmentation network using mini-batch stochastic gradient descent with batch size 16, momentum 0.9, and weight decay 0.0005. The initial learning rate is set to 0.01, the exponential decay method is used to control the learning rate of each iteration, and the power is set to 0.9. This method is implemented on the PyTorch platform. The machine uses an NVIDIA GeForce GTX 1080 Ti for experiments.

**Table 1.** The fusion augmentation training process.

| **Joint Low-Level and High-Level Training Strategy** |
| --- |
| Input: infrared images and visible images |
| Output: fused images |
| For m ≤ M: |
| for p steps: |
|     select b infrared images; select b visible images |
|     update the weight of semantic loss β |
|     update the parameters of the fusion network Nf by optimizer |
|   generate fused images |
|   for q steps: |
|     select b fused images |
|     update the parameters of the segmentation network Ns by optimizer |

**Table 2.** The GAN augmentation training process.

| **Generative Adversarial Networks Training Strategy** |
|---|
| Input: fused images (adversarial training iterations time is T, training iteration of the discriminant network is K, small batch sample size is M) |
| Output: generator G (z, theta) |
| 1 Random initialization |
| 2 for t tend to 1 to T do: |
|     For k tend to 1 to K do: |
|       Select M samples $\left\{x^{(m)}\right\}, 1 \leq m \leq M$ from training set D |
|       Select M samples $\left\{z^{(m)}\right\}, 1 \leq m \leq M$ from distribution N(0,1) |
|       Update φ using random gradient ascending, gradient is |
|       $\frac{\partial}{\partial\phi}\left[\frac{1}{M}\sum\limits_{m=1}^{M}\left(\log D\left(x^{(m)},\phi\right)+\log\left(1-D\left(G\left(z^{(m)},\theta\right),\phi\right)\right)\right)\right]$ |
|     End |
|     Select M samples $\left\{z^{(m)}\right\}, 1 \leq m \leq M$ from distribution N(0,1) |
|     Update theta using random gradient ascending, gradient is |
|     $\frac{\partial}{\partial\phi}\left[\frac{1}{M}\sum\limits_{m=1}^{M}D\left(G\left(z^{(m)},\theta\right),\phi\right)\right]$ |
|     end |

### 4.2.2. Residual Convolutional GAN Augmentation Training

The GAN network improved by Resblock can generate new images with similar distribution according to the fused images, thus expanding the target detection dataset in this paper and improving the generalization ability and robustness of the model. The network is referred to as RCGAN [24] in this paper.

The training process is in Table 2.

In the process, x is the data representation of the image. D (x) represents the possibility that the discriminator judges x comes from the real training data. z is a latent space vector sampled from the standard normal distribution [25]. G (z) represents the function of the generator mapping from the latent vector z to the data space. The purpose of G is to estimate the distribution from the training data. D (G(z)) is the probability that the output of generator G is a real image.

The discriminator tries to maximize the probability of correctly distinguishing between true and false (logD (x)), and the generator tries to minimize the probability that D predicts the output of G to be false log( 1−D (G(x))).

Training images are scaled to the range of the tanh activation function [−1, 1]. The models are trained with mini-batch stochastic gradient descent (SGD) with a mini-batch size of 128. All weights are initialized from a zero-centered normal distribution with standard deviation 0.02. In the LeakyReLU, the slope of the leak was set to 0.2. We used the Adam optimizer with tuned hyperparameters. Additionally, we used the learning rate of 0.002.

### 4.3. Performance Comparison

Firstly (step one), images are enhanced by fusing visible and infrared images, which belongs to multi-sample augmentation. It is obvious that the data augmentation operation overcomes the disadvantages of visible images. From Figure 9, pedestrians with temperature information can be captured by infrared cameras at night, gaining the advantage of infrared information and enhancing the data.

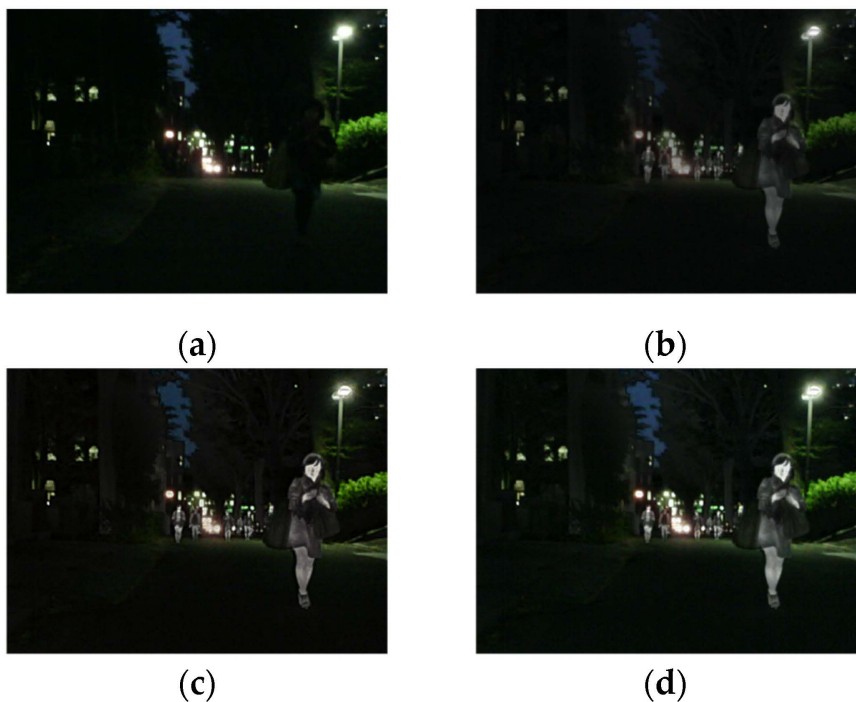

**Figure 9.** (**a**) a visible original image; (**b**) fused image using Densefuse; (**c**) fused image using Lpfuse; (**d**) fused image using our methods.

In Figure 9, the same infrared image is fused with (a), and the results of three different fusion methods are shown in (b), (c), and (d). It can be seen that infrared and visible fusion has an obvious effect, otherwise, the pedestrian would not be found. Moreover, our improved multi-sample data augmentation perform best in the comparison, the quantitative results are in the Table 3.

**Table 3.** The detection effect comparison of multi-sample data augmented images and original images at night.

| Methods | P | R | mAP@0.5 |
|---|---|---|---|
| Visible original image | 0.66 | 0.174 | 0.209 |
| Densefuse | 0.831 | 0.679 | 0.706 |
| Lpfuse | 0.64 | 0.756 | 0.642 |
| Ours | 0.786 | 0.795 | 0.76 |

Table 1 indicates that it is correct to introduce infrared images to poor-light images, and our fusion methods obtains better performance in the pedestrian detection task. The improved fusion method better enhances the data, so that the average detection accuracy ranks first among the four situations.

Secondly (step two), 50 different fused images from the previous step are put in the Residual Convolutional GAN to generate 500 fake images. Some of the new samples are depicted in Figure 10. From the picture, it is seen that the people in the fake images can be detected in most cases. We can judge the authenticity of the results by the shape of the larger human form.

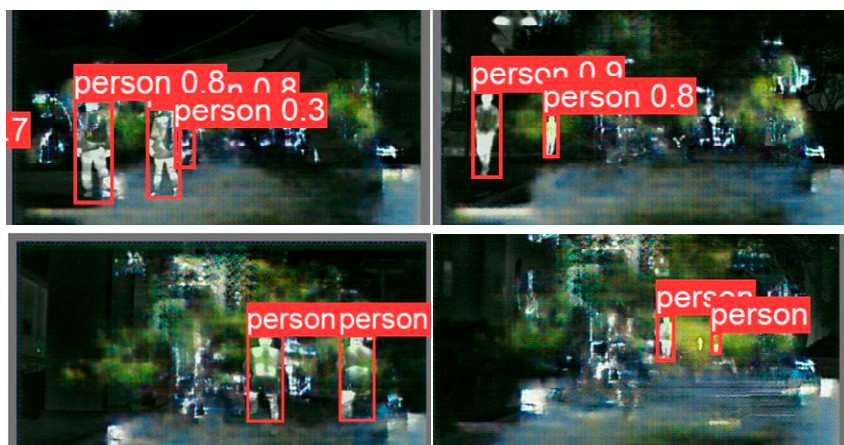

**Figure 10.** New generated images by RCGAN network.

The generated fake images have the same distribution as the input images, and the objects in them can be found by detection frames. The batch of images are good resources for data augmentation.

Finally, the Retinex deillumination method is applied on the augmented dataset, the effect is seen in Figure 11. Intuitively, the pictures after simple single-sample enhancement obtained higher definition and recognition. The ablation study is conducted. We removed the deillumination block and observed the effect of detection. The precision fell by 0.124.

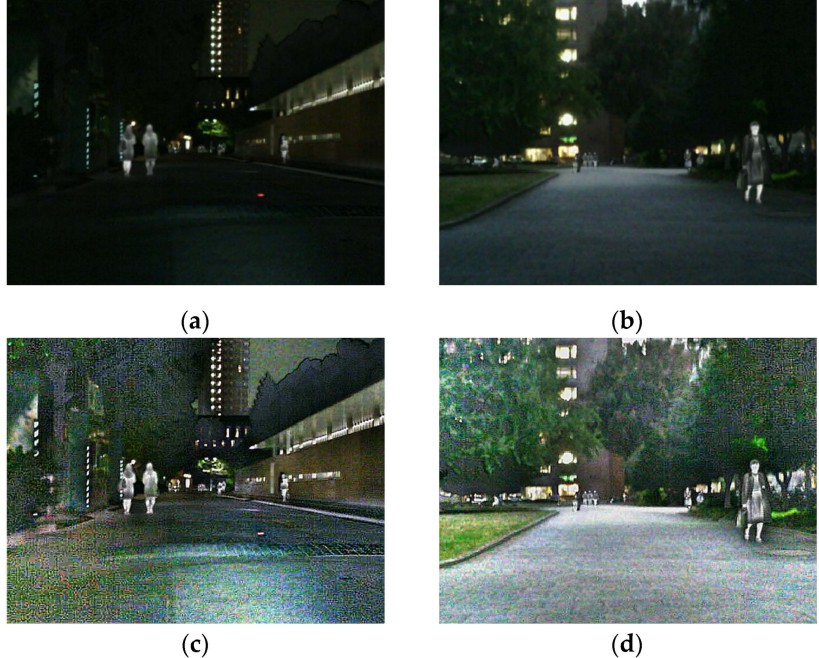

**Figure 11.** (**a**,**b**): random images from first two steps; (**c**): (**a**) after Retinex deillumination augmentation; (**d**): (**b**) after Retinex deillumination augmentation.

The overall experimental results are listed in Table 4. This paper performs a comparison between three data augmentation methods. The detection accuracy mAP of the model trained on the fusion-enhanced dataset is 0.76. The detection model is trained using the augmented dataset after fusion enhancement and GAN enhancement, the detection effect mAP is 0.854. At the same time, both the precision and recall rate of pedestrian detection are improved. With all the three augmentation methods, the metrics of mAP reach 0.97. Meanwhile, we calculate the floating point operations per second (GFLOPs). Models trained on datasets expanded in different ways have different amount of computation

results. Overall, the computational complexity of the improved model is average and the speed is good.

**Table 4.** The overall comparison results for pedestrian detection.

| Image | Method | P | R | mAP@0.5 | GFLOPs |
|---|---|---|---|---|---|
| Original visible images | No addition | 0.66 | 0.174 | 0.209 | 15.8 |
| Augmented images | +fusion | 0.786 | 0.795 | 0.76 | 15.9 |
| | +fusion +GAN | 0.869 | 0.833 | 0.854 | 17.2 |
| | +fusion +GAN +deillumination | 0.955 | 0.896 | 0.97 | 14.1 |

Figure 12 shows the detect effect on the augmented images by our methods. Although nighttime images are difficult to discern, data augmentation can greatly improve the model's detection ability.

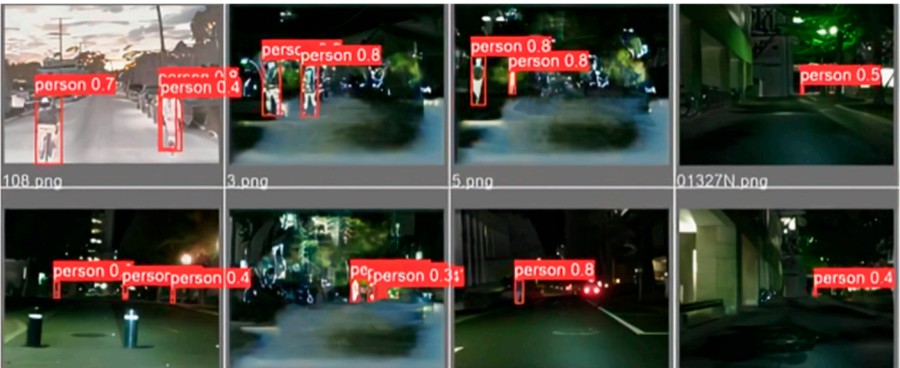

**Figure 12.** The detect effect on the augmented images by our methods.

In Figure 13, the left nine images are original, and the right nine are enhanced images. After GAN augmentation, more useful objects emerged in the pictures. More people were seen in the dark. The comparison of the F1 score and confusion matrix are shown in the Figure 14.

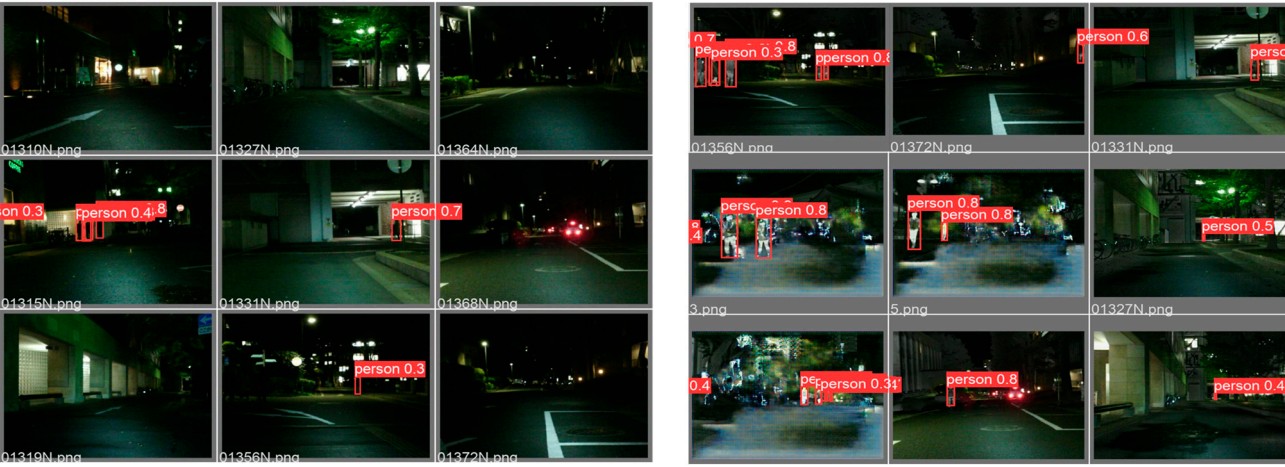

**Figure 13.** The visible image at night and the augmentation result.

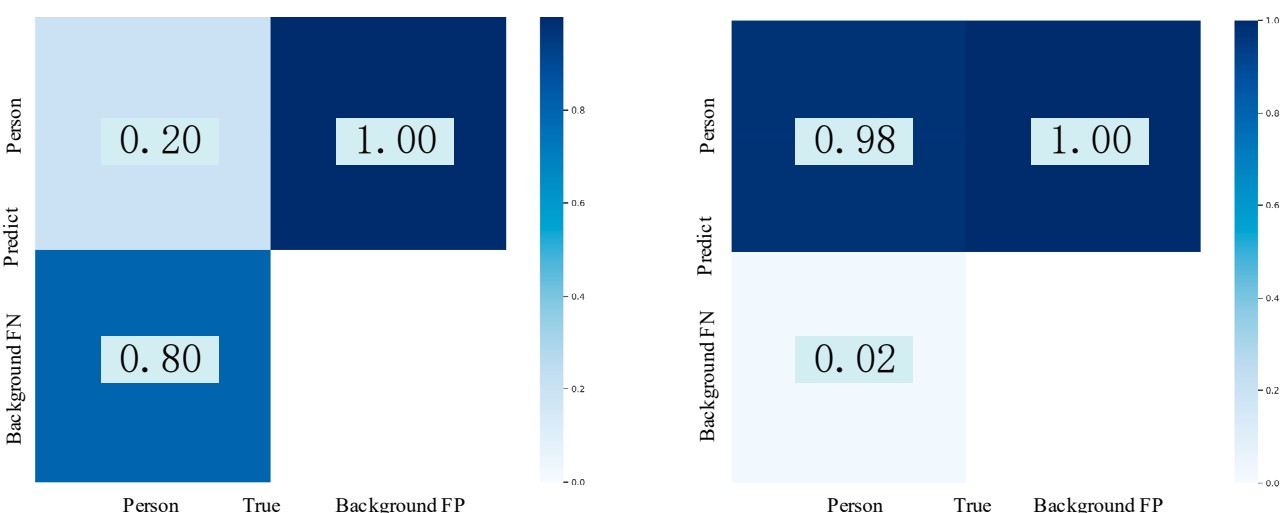

**Figure 14.** The confusion matrixes before and after the enhancement.

Due to the few samples are tested, the gap is obvious in Figure 14. It is also proved that the multi-sample augmentation and GAN make contributions to the detection effects.

*4.4. Application*

Through drone sampling, we obtained the custom dataset in Figure 15, and we used our methods on the dataset. The results show that the methods have a certain universality. The RCGAN is applied to the dataset to increase the number of datasets. The fake dataset is shown in Figure 16. The detection result is displayed in Figure 17. It is apparent that the detection model trained by the enhanced dataset has higher precision and robustness. Figure 18 shows the detection P-R curve after GAN augmentation, mAP is 0.977, bigger than 0.97 in the Table 4.

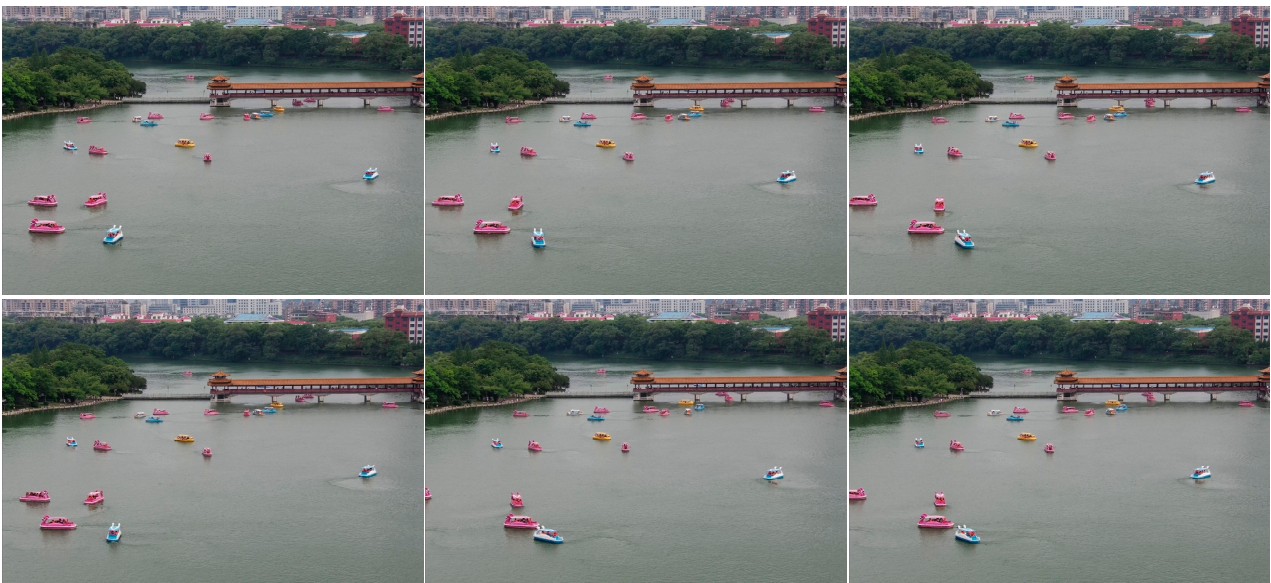

**Figure 15.** The custom dataset.

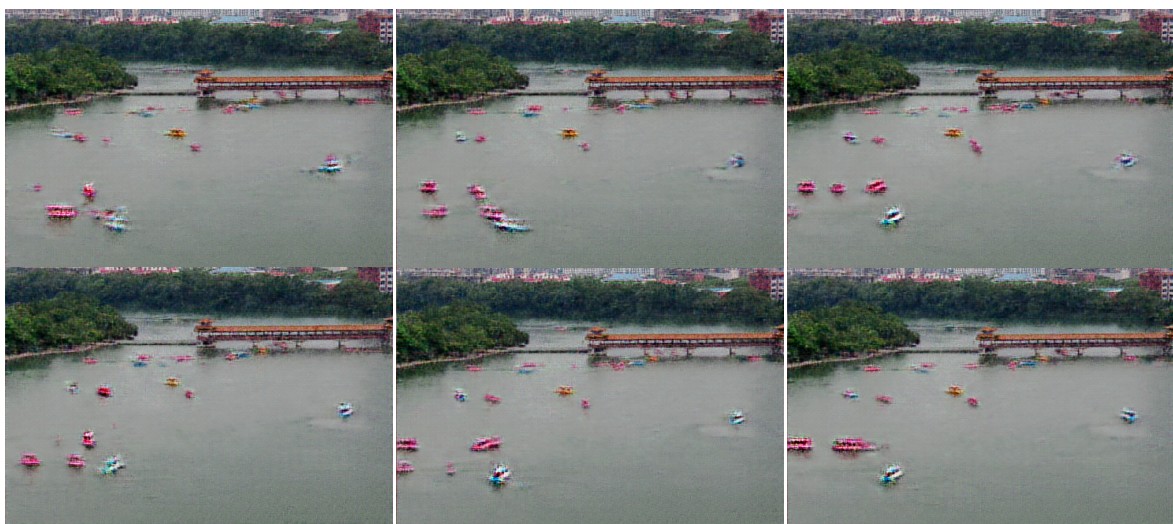

**Figure 16.** The fake dataset.

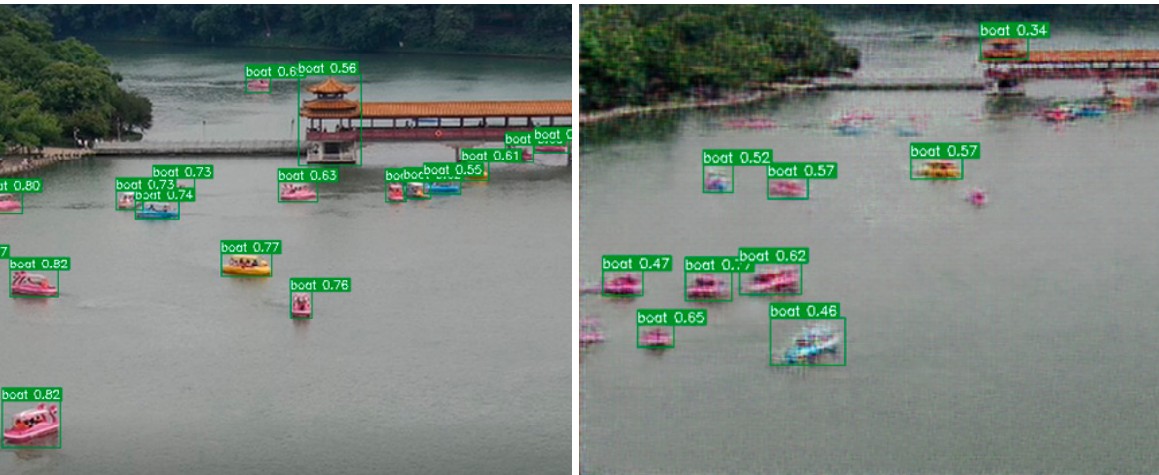

**Figure 17.** Detection results displayed after the enhanced model.

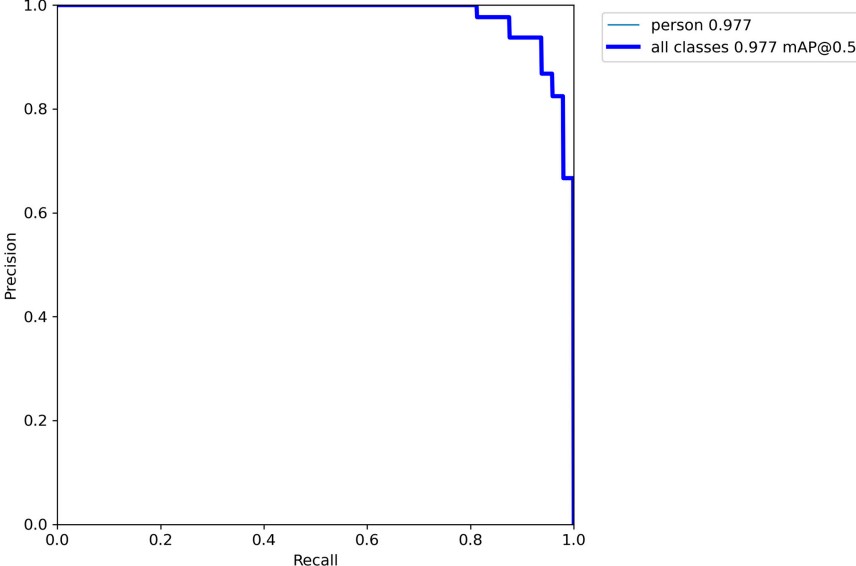

**Figure 18.** Detection result displayed after the enhanced model.

## 5. Conclusions

For images acquired under inadequate conditions, it is necessary to perform data augmentation. In this paper, we proposed improved data augmentation methods and tested their pedestrian detection methods on the MSRS dataset. Then, we compared the impact of supervised multi-sample fusion augmentation, single-sample deillumination augmentation, and unsupervised RCGAN augmentation on target detection. This paper improves the bilateral attention semantic segmentation network in visible light and infrared fusion, which enhances the representation ability of the convolution network and makes the target detection task better to drive multi-sample fusion augmentation. On the basis of DCGAN, the residual convolution module is added to generate more realistic and smooth new images to expand the dataset. For a single sample, the deillumination and sharpening augmentation can greatly improve the image clarity. It can be seen that their effects are different, but they both effectively improved image quality and played a good role in promoting pedestrian detection.

There are some limitations of the study, however. In multi-sample enhancement, it is necessary to use a registered pair of infrared and visible light images. If it is an image with different focal lengths, it needs to be registered first; since the pedestrian target in the dark image is small and the background is strongly nonlinear, the background of the generated image of GAN is relatively abstract. In future research, we hope to implement a lightweight image preprocessing module to solve the problem of rapid image registration and improve the generative adversarial network to make the background of the resulting image more realistic.

**Author Contributions:** Conceptualization, B.L. and S.S.; methodology, J.W.; software, B.L.; validation, B.L., J.W. and S.S.; formal analysis, B.L.; investigation, B.L.; resources, B.L.; data curation, J.W.; writing—original draft preparation, B.L.; writing—review and editing, B.L.; visualization, B.L.; supervision, S.S.; project administration, S.S.; funding acquisition, J.W. All authors have read and agreed to the published version of the manuscript.

**Funding:** This research is financially supported by the National Natural Youth Science Foundation of China under Grant No. 62201598. This work is supported by School of Intelligent Science of National University of Defense Technology.

**Data Availability Statement:** The dataset (MSRS) used in this paper is publicly available and can be download from https://github.com/Linfeng-Tang/MSRS (accessed on 26 September 2022).

**Conflicts of Interest:** The authors declare no conflict of interest.

## Abbreviations

| | |
|---|---|
| GAN | Generative Adversarial Network |
| mAP | Mean Average Precision |
| DCGAN | Deep Convolution Generative Adversarial Network |
| GAP | Global Average Pooling |
| GCT | Gated Channel Attention Mechanism |
| CSP | Cross Stage Partial |
| FPN | Feature Pyramid Networks |
| RDG | Residual Dense Gradient Block |

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
