# Peer review of "The Effect of Data Augmentation Methods on Pedestrian Object Detection"

_electronics, doi:10.3390/electronics11193185_

Round 1

Reviewer 1 Report

The comments are attached as PDF. Implement them to improve the quality of the manuscript.

Reviewer 2 Report

- The abstract should be improved, the initial background is confusing.

- The section 1.1 is shallow, the authors should discuss more some aspects of: pedestrian detection problem and why data augmentation is needed in deep learning. 

- The section 1.2, should be separated from the introduction into a Related works section.

- The section 2.2 could be reduced, there is no need to explain deeply Yolo vs EfficientDet for example, Figure 4 is irrelevant to the current paper.

- In section 4.1, the authors selected "poor lightning" images, how were them selected? Visually?

- The conclusion should be improved, the authors should be discuss details regarding study limitations and future works.

Reviewer 3 Report

The paper titled "The Effect of data augmentation methods on pedestrian object detection" discusses the impact of several data augmentation methods to achieve pedestrian detection. For the image data collected at night un-der limited conditions, three different types of enhancement methods are used to verify whether they can promote the final pedestrian detection.

My few suggestions are listed below.

1. At the end of Introduction, authors should introduce a table that should contain the Acronyms used in the paper. Currently, the abstract of the manuscript also contain few acronyms that should be minimised from the abstract.

2. At the end of Section 1, authors have listed limitations of the current methods. I propose to authors to write their contributions clearly at the end of Section 1.

3. In Fig. 4 authors should should use different patterns of line to be clearly shown in print version. Currently, in print version, all lines appeared similar.

4. Section 3 is bit lengthy. I propose to authors to include a pseudo code of their developed method. This will make the paper easy to understand.

5. Although authors report higher mAP values of their proposed method than few of the compared methods. However, computational complexity is still a question. I propose to authors to include the computational complexity of your method on various image resolutions, then list the computational comparison.

6. In Section 5, few sentences should be added to hint towards the possible future works.

7. Quality and number of references should be improved. I propose to authors to include few latest references from the years 2021 and onwards.

8. Authors should also include ablation study of their proposed method.

Round 2

Reviewer 1 Report

The comments are attached as PDF file
